Genome-wide identification, classification, and expression analysis of the HSF gene family in pineapple (Ananas comosus)

Wang Lulu 1
Liu Yanhui 1
Chai Mengnan 1
Chen Huihuang 1
Aslam Mohammad 1
Niu Xiaoping 2
Qin Yuan yuanqin@fafu.edu.cn 1 2
Cai Hanyang caihanyang123@163.com 1
1 State Key Lab of Ecological Pest Control for Fujian and Taiwan Crops; Key Lab of Genetics, Breeding and Multiple Utilization of Crops, Ministry of Education; Fujian Provincial Key Lab of Haixia Applied Plant Systems Biology, College of Life Sciences, Fuji , Fuzhou , Fujian , China
2 State Key Laboratory for Conservation and Utilization of Subtropical Agro-Bioresources, Guangxi Key Lab of Sugarcane Biology, College of Agriculture, Guangxi University , Nanning , Guangxi , China
Adhikari Tika
Electronic publication date: 2021 Apr 27
Publication date: 2021
Volume: 9
Electronic Location ID: e11329
Received 2020 Dec 16; Accepted 2021 Mar 31
Copyright: ©2021 Wang et al.
Copyright year: 2021
Copyright holder: Wang et al.
License: This is an open access article distributed under the terms of the Creative Commons Attribution License, which permits unrestricted use, distribution, reproduction and adaptation in any medium and for any purpose provided that it is properly attributed. For attribution, the original author(s), title, publication source (PeerJ) and either DOI or URL of the article must be cited.
License URL: https://creativecommons.org/licenses/by/4.0/

Keywords: AcHSF, Phylogenetic analysis, Pineapple, Cold, Heat, ABA

Funding: National Natural Science Foundation of China U1605212 31761130074 31970333 31700279 Guangxi Distinguished Experts Fellowship This work was supported by the National Natural Science Foundation of China (U1605212, 31761130074 and 31970333 to Yuan Qin; 31700279 to Hanyang Cai) and a Guangxi Distinguished Experts Fellowship to Yuan Qin. The funders had no role in study design, data collection and analysis, decision to publish, or preparation of the manuscript.

==============================
Transcription factors (TFs), such as heat shock transcription factors (HSFs), usually play critical regulatory functions in plant development, growth, and response to environmental cues. However, no HSFs have been characterized in pineapple thus far. Here, we identified 22 AcHSF genes from the pineapple genome. Gene structure, motifs, and phylogenetic analysis showed that AcHSF families were distinctly grouped into three subfamilies (12 in Group A, seven in Group B, and four in Group C). The AcHSF promoters contained various cis-elements associated with stress, hormones, and plant development processes, for instance, STRE, WRKY, and ABRE binding sites. The majority of HSFs were expressed in diverse pineapple tissues and developmental stages. The expression of AcHSF-B4b/AcHSF-B4c and AcHSF-A7b/AcHSF-A1c were enriched in the ovules and fruits, respectively. Six genes (AcHSF-A1a , AcHSF-A2, AcHSF-A9a, AcHSF-B1a, AcHSF-B2a, and AcHSF-C1a) were transcriptionally modified by cold, heat, and ABA. Our results provide an overview and lay the foundation for future functional characterization of the pineapple HSF gene family.

Introduction

The living environment of plants is faced with many challenges, including cold, heat, drought, and salinity stresses (Hu & Xiong, 2014; Pereira, 2016; Zhu, 2016). Due to global warming, heat stress is becoming a serious agricultural threat for agricultural production and planting areas worldwide (Wahid et al., 2007). Typically, plants face heat stress when the temperature rises 10 to 15 degrees above the optimum growth environment. Heat stress affects plant development and growth and eventually leads to a decrease in crop yield. Consequently, as a defense or signaling mechanism in response to environmental stresses, plants regulate the expression of several genes through different transcription factors (TFs). The heat shock transcription factor family of plants is involved in heat stress response and regulates the expression of several stress-responsive proteins, including heat shock proteins (HSPs), ascorbate peroxidase (APX), and catalase (CAT) (Ohama et al., 2017). Previously, several studies have validated the roles of heat shock transcription factors (HSFs) in plant stress response. For example, the overexpression of CarHsf-A2 could enhance chickpea stress tolerance without any pleiotropic effects (Chidambaranathan et al., 2018). Besides, AtHSF-B1 and AtHsf-B2b act as expression repressors after heat-stress, and the AtHSF-A1 is involved in cold acclimation in Arabidopsis thaliana (Ikeda, Mitsuda & Ohme-Takagi, 2011; Olate et al., 2018). Similarly, PeuHsf-A2 gets induced by heat stress, increasing desert poplar acclimation (Zhang et al., 2016b).

HSFs contain several evolutionarily conserved functional domains. The canonical HSF protein contains an N-terminal DNA-binding domain (DBD) that binds to HSEs; a hydrophobic amino acid residue (HR-A/B) oligomerization domain (OD) heptad repeat bound to DBD by a flexible linker. In addition, it also includes a region of nuclear localization signal (NLS), a region of nuclear export signal (NES) and a motif of activator AHA located at the C-terminal (Guo et al., 2016; Nishizawa-Yokoi et al., 2011; Singh et al., 2012; Yabuta, 2016). Three types of plant HSF have been identified based on the variable linker (usually 15–80 aa) and HR-A/B domain, including nine class A (A1–A9), four class B (B1–B4), and two class C (C1–C2) (Giesguth et al., 2015; Nishizawa-Yokoi et al., 2011; Shim et al., 2009; Singh et al., 2012; Yabuta, 2016). Initially, HSFs were identified in yeast (Scharf et al., 1990), and the first HSF plant gene was identified in tomatoes (Sorger& Pelham, 1988). Since then, several plant HSF gene families, including Arabidopsis thaliana (Guo et al., 2008), rice (Oryza sativa L.) (Chauhan et al., 2011; Jin, Gho & Jung, 2013), maize (Zea mays L.) (Lin et al., 2014), Populus trichocarpa (Wang et al., 2012), wheat (Triticum aestivum L.) (Chauhan et al., 2013), soybean (Glycine max) (Chung, Kim & Lee, 2013), Chinese cabbage (Brassica rapa ssp. pekinensis) (Song et al., 2014), cotton (Gossypium hirsutum) (Wang et al., 2014), barrel medic (Medicago truncatula) (Lin et al., 2014), pepper (Capsicum annuum L.) (Guo et al., 2015; Guo et al., 2014), strawberry (Fragaria vesca) (Hu et al., 2015) and tea plant (Camellia sinensis) (Liu et al., 2016), etc.

Pineapple (Ananas comosus) is grown in subtropical and tropical regions and is widely loved worldwide and its genome is sequenced (Bai et al., 2019; Ming et al., 2015). However, many biotic and abiotic environmental stresses, pathogen infection, and degradation of good breeds limit pineapple production (Barral et al., 2019). Therefore, it is essential to identify and characterize genes involved in response to environmental stresses and study the underlying molecular mechanism that could be used for possible genetic breeding applications. HSFs are well known for the co-involvement in different environmental cues, such as cold, heat, and ABA. However, AcHSFs and their possible role in pineapples have not been explored. Here, whole-genome identification and expression analysis of AcHSFs gene during flower, fruit development, and abiotic stress was conducted to expand the understanding of AcHSFs gene and its application in genetic breeding.

Methods

Identification and Characterization of AcHSF genes in pineapple

The amino acid sequences of pineapple HSFs were retrieved using the HSF-type DBD domain (Pfam: PF00447) as a query in Phytozome JGI. A total of 30 pineapple HSFs were obtained from JGI. Only 22 AcHSFs left, after filtering out the non-typical HSF-type DBD and repeated sequences or canonical coiled-coil structures by SMART online tool (Letunic, Doerks & Bork, 2012). The information of Chromosome localization, CDS, and AA length for AcHSFs was obtained from JGI Phytozome v12.1. The biophysical properties of coding AcHSFs were calculated using the Expasy ProtParam tool. The subcellular localization of AcHSFs was analyzed using BUSCA (http://busca.biocomp.unibo.it/).

Chromosome Localization phylogenetic relationships

The information of all pineapple HSFs’ chromosome localization site was acquired from Phytozome v12.1, including chromosome length, chromosome location, and gene start site. The MapChart v2.0 (https://mapchart.net/) was adopted to map the chromosomal location. The phylogenetic relationship of different HSF proteins was explored, and the phylogenetic tree was created using the AcHSF amino acid sequences and the other three species, Arabidopsis thaliana, A. thaliana, Oryza sativa (O. sativa), and Populus. trichocarpa (P. trichocarpa) by the MEGAX with a bootstrap value of 1000. The HSF gene in pineapples was referred to as AcHSF genes and classified according to HSFs in the phylogenetic tree classes A, B, and C.

Genetic structure and cis-acting elements

The gene structures of AcHSF, including exons, introns, and UTR were displayed by the GSDS online tool (Guo et al., 2007). The promoter sequence of AcHSFs found in Phytozome is located 2kb upstream of the translation initiation site. These sequences were analyzed using a plant cis-acting element database New PLACE (Higo et al., 1999), to identify cis-elements necessary for gene expression, development, and hormone signaling under abiotic stress.

Conserved domains and motifs analysis of AcHSFs

Clustal X 2.0 and DNAMAN software was used to align and edit the DBD domain and HR-A/B regions (OD). Using the cNLS online tool, NLS domains were predicted and the NetNES 1.1 online server identified NES domains in the AcHSFs. MEME server (http://meme-suite.org/) was applied to define the conserved motifs of AcHSFs following the parameters: the number of repetitions = any, the maximum number of motifs = 10, minimum width ≥10, maximum width ≤50, and motifs with an E-value <0.01 were retained.

Expression patterns analysis

The transcriptomic data generated from different organs and developmental stages of pineapple have been described previously (Ming et al., 2015; Wang et al., 2020). Briefly, the different organs include 3 stages of petal tissues, 4 stages of sepal tissues, 6 stages of stamen tissues, 7 stages of ovule tissues, 7 stages of gynoecium tissues, and root, mixed flower, leaf, and 6 stages of fruit were used to generate heatmap using the pheatmap package of R software.

Stress treatments

One-month-old uniform tissue-cultured seedlings in a rooting medium were used for the stress treatment analyses (Priyadarshani et al., 2018). The stress treatments of pineapple seeding as follows: cold (4 ° C), heat (45 ° C), and ABA (100 mM). Leaves were collected from three independent plants at 12 h, 24 h and 48 h after treatment and immediately stored in liquid nitrogen before RNA extraction.Untreated pineapple seedlings of the same size were used as controls.

RNA extraction and qRT-PCR

Total RNA of pineapple leaf tissues was obtained using the RNeasy Plant Mini Kit (50) (Qiagen). For cDNA synthesis, a total of 1 µg RNA per sample was used with cDNA Synthesis SuperMix (Transgen, Beijing, China). qPCR was conducted by the Bio-Rad CFX manager machine using TransStart® Top Green qPCR SuperMix (Transgen, Beijing, China). Pineapple Actin2 is used as the reference gene for qPCR (Wang et al., 2020). There are total of three biological replicates for each sample, and the results are shown as the mean ± standard deviations.

Results

Genome-wide identification of HSF genes in pineapple

HSF-type DBD domains (Pfam: PF00447) amino acid sequences were submitted into Ananas comosus v3 Phytozome database v12.1. A total of 30 putative pineapple HSF sequences were acquired. And then, checked by the SMART online tool and Pfam database, 1 pineapple HSF sequence was rejected due to the absence of typical HSF-DBD domains, and 7 HSF sequences were abandoned due to the absence of coiled-coil structure. As a result, 22 non-redundant pineapple HSFs were identified (Table 1). The comprehensive information of these 22 AcHSF genes, including gene name, gene ID, CDS and protein length, isoelectric points, molecular weights, predicted subcellular location, and other features are presented in Table 1. The gene with the longest amino acid length is ACHSF-A5, which contains 601 amino acids and also has the largest molecular weight of 65.81 KDa; and the gene with the shortest amino acid length was AcHSF-A9b, which contains 129 amino acids and also has the smallest molecular weight of 13.77 KDa. Prediction of protein isoelectric points (pI) can aid in the purification and isolation of proteins. The predicted isoelectric points (pI) of AcHSFs ranged from 4.68 (AcHSF-B2b) to 9.63 (AcHSF-B4c). Detailed information on other parameters has been given in Table 1.

Table 1 Protein information of pineapple (Ananas comosus) heat shock transcription factors (AcHSFs).

Including protein name, sequenced ID, subfamily, Chromosome Localization, CDS and amino acid (AA) length, molecular weight (MW), isoelectric point (pI), and predicted subcellular location.

NO	Protein name	Gene ID	Subfamily	Chromosome Localization	CDS length	AA length	pI	Mw (kDa)	Predicted subcellular location	
1	AcHSF-A1a	Aco027746.1	A1	Chr 5	11287848-11300490	1752	583	5.08	63.91	nucleus	
2	AcHSF-A1b	Aco020850.1	A1	Chr 9	8088573-8101648	1725	574	4.86	62.37	nucleus	
3	AcHSF-A1c	Aco016980.1	A1	Chr 8	13365954-13368969	1338	445	4.89	48.59	nucleus	
4	AcHSF-A2	Aco008819.1	A2	Chr 9	2742350-2746933	1098	365	5.44	41.31	nucleus	
5	AcHSF-A3	Aco016689.1	A3	Chr 17	159803-163500	1671	556	5.03	61.08	nucleus	
6	AcHSF-A4	Aco005592.1	A4	Chr 11	11506920-11509712	1359	452	5.28	51.19	nucleus	
7	AcHSF-A5	Aco016114.1	A5	Chr 21	1487000-1489970	1806	601	7.68	65.81	nucleus	
8	AcHSF-A6	Aco009685.1	A6	Ch r1	380459-383347	1116	371	4.78	41.94	nucleus	
9	AcHSF-A7a	Aco005862.1	A7	Chr 16	10841121-10844044	1089	362	5.8	41.47	nucleus	
10	AcHSF-A7b	Aco015210.1	A7	Chr 5	1225625-1227632	999	332	4.8	36.96	nucleus	
11	AcHSF-A9a	Aco021879.1	A9	Chr 21	5356086-5390683	1395	464	5.17	51.49	nucleus	
12	AcHSF-A9b	Aco022474.1	A9	Chr 1	5983677-5984066	390	129	5.18	13.77	nucleus	
14	AcHSF-B1a	Aco002688.1	B1	Chr 6	10679048-10683679	948	315	8.54	34.28	nucleus	
13	AcHSF-B1b	Aco030273.1	B1	scaffold_1756	6908-10841	930	309	8.72	33.57	nucleus	
15	AcHSF-B1c	Aco001320.1	B1	Chr 2	12375558-12379157	930	309	8.88	33.76	nucleus	
16	AcHSF-B2a	Aco013873.1	B2	Chr 8	9295304-9296338	906	301	6.06	32.77	nucleus	
17	AcHSF-B2b	Aco027680.1	B2	scaffold_382	177006-178278	1173	390	4.68	40.94	nucleus	
18	AcHSF-B4b	Aco017163.1	B4	Chr 18	739089-740863	957	318	5.7	36.55	nucleus	
19	AcHSF-B4c	Aco031324.1	B4	scaffold_2094	3028-3903	876	291	9.63	31.26	extracellular space	
20	AcHSF-C1a	Aco006046.1	C1	Chr 16	9444927-9446130	888	295	6.35	33.69	nucleus	
21	AcHSF-C1b	Aco005573.1	C1	Chr 11	11371568-11373537	696	231	8.52	26.42	nucleus	
22	AcHSF-C2	Aco027352.1	C2	Chr 5	1814813-1816318	1113	370	8.3	39.62	nucleus	

According to the detailed gene information, 19 AcHSF genes were mapped to the 11 pineapple chromosomes and 3 AcHSF genes located in the scaffold (Table 1). The number of pineapple HSF genes for each chromosome varied significantly, and there is no discernible pattern in the location of these genes on chromosomes. For example, three AcHSF genes were located in chromosome 5, whereas only one was present in chromosomes 2, 6, 17, and 18 respectively (Fig. 1).

Figure 1 Distribution of AcHSF genes in pineapple genome.

The different color bars represent the different chromosomes and the chromosome numbers are showed on the top of the bars. Length of the bars are related with the size of the chromosomes. AcHSF genes distribute on the 11 chromosomes. The white bar represent the scale bar 25 Mb.

Figure 2 The phylogenetic tree of HSF proteins.

The phylogenetic tree of HSF proteins in pineapple and other plant species was generated by MEGA 5 using the neighbor-joining method and the bootstrap values were set at 1000. AcHSFs were divided into three classes and 13 subclasses (A1, A2, A3, A4, A5, A6, A7, A9, B1, B2, B4, C1 and C2) and separated by green dots. Ac, Ananas comosus; At, Arabidopsis thaliana; Os, Oryza sativa and Pt, Populus trichocarpa.

Phylogenetic analysis of AcHSFs gene family

A phylogenetic analysis of 31 Populus trichocarpa HSFs, 25 rice HSFs, and 21 Arabidopsis HSFs was performed to classify the phylogenetic relationships (Guo et al., 2016), together with those of AcHSFs by generating a neighbor-joining phylogenetic tree. The HSFs were grouped into three clusters, A, B, and C, according to the difference between the amino acid sequences of the DBD domain, the HR-A/B region, and the linker between them (Guo et al., 2016; Scharf et al., 2012). Class A consisted of 10 subclusters, designated A1 to A10. Class B contained B1 to B4, and Class C comprises C1 and C2 sub-clusters. In pineapple (Ananas comosus), according to their phylogenetic relationship, 12 AcHSFs out of 22 proteins belong to class A, followed by seven AcHSFs belonging to class B, and three copies of class C (Fig. 2). As a monocot, the pineapple was more similar to rice, rather than the dicot Arabidopsis and Populus trichocarpa. However, none of the AcHSFs were found in the subclass A8 and B3, which was reported to only exist in the monocots (Li et al., 2014). It is strange that the pineapple and rice subclass A7 HSFs showed higher similarity to A2 rather than the Arabidopsis and Populus trichocarpa subclass A7, and the AtHSF-A6a also shows abnormal clustering (Fig. 2).

Gene structures and cis -acting elements analysis of AcHSFs

Intron, exon, 5′ UTR, and 3′ UTR structures were analyzed using Gene Structure Display Server (GSDS) v2.0.0 to reveal the gene structural features of AcHSFs. The number of exons for AcHSFs ranged from 1 to 5 (Fig. 3), while only in the longest AcHSF-A9a (genomic sequence 34,598 bp.) was found 5 exons. The 5′ UTR, and 3′ UTR sequence of the AcHSF genes are incomplete, 8 out of 22 AcHSFs (AcHSF-A3, AcHSF-A7b, AcHSF-B1c, AcHSF-B2a, AcHSF-B2b, AcHSF-B4c, AcHSF-C1a, and AcHSF-C1b) do not have 5′ UTR and 3′ UTR sequences, 3 AcHSFs (AcHSF-A1b, AcHSF-A9a, and AcHSF-B4b) have only 5′UTR sequences, 6 AcHSFs (AcHSF-A4, AcHSF-A5, AcHSF-A7, AcHSF-B1a, AcHSF-B1b, and AcHSF-C2) have only 3′UTR sequences, and 5 AcHSFs (AcHSF-A1a, AcHSF-A1c, AcHSF-A2, AcHSF-A6, and AcHSF-A9) both have 5′ UTR and 3′ UTR sequences.

Figure 3 Exon–intron organization of AcHSF genes.

The exons, introns and untranslated regions (UTRs) were indicated by the yellow boxes, black lines and blue boxes, respectively.

Stress responses elements of the dehydration-responsive element (DRE), ABA-responsive element (ABRE), low-temperature responsive element (LTRE), MYB, MYC, and WRKY elements has been reported to play important roles in drought, salt, cold, ABA, and GA responses (Chai et al., 2020; Li et al., 2012; Zhang et al., 2009). The 2kb sequences upstream of AcHSFs gene were selected for analysis. The cis-acting elements analysis of AcHSFs promoter demonstrated that every pineapple HSF contains at least 2 MYB, MYC, and WRKY elements, except for AcHSF-B4c (Table 2). But for the AcHSF-B4c, only 110bp promoter sequence can be found in the upstream area, among the 110bp promoter sequence, the main core component of the promoter TATA-box and CAAT-box, the light regulatory element (RYREPEATBNNAPA), and root-specific expression related elements (ROOTMOTIFTAPOX1) can be found. We also detected the ABRE, DRE, and LTRE in the AcHSFs promoter area. The result showed that the AcHSF-A1a and AcHSF-A4 lacked ABRE, the AcHSF-A6 and AcHSF-C1b lacked DRE, the AcHSF-C1b lacked LTRE, and the AcHSF-A9a, AcHSF-B4b, and AcHSF-B4c did not have these three stress response elements (Table 2). The cis-element studies in the promoters indicate that HSFs are highly related to the response to stress.

Table 2 Distribution of ABRE, DRE, LTRE, MYB, MYC and WRKY cis-acting elements in pineapple HSF promoters.

Gene	ABRE	DRE	LTRE	MYB	MYC	WRKY	
AcHSF-A1a	0	1	5	10	2	10	
AcHSF-A1b	1	1	5	7	2	8	
AcHSF-A1c	14	2	2	2	2	2	
AcHSF-A2	12	4	6	17	4	7	
AcHSF-A3	11	4	6	22	10	6	
AcHSF-A4	0	2	4	18	12	10	
AcHSF-A5	8	1	4	19	20	7	
AcHSF-A6	1	0	2	15	8	3	
AcHSF-A7a	2	1	5	24	22	7	
AcHSF-A7b	12	2	3	20	8	11	
AcHSF-A9a	0	0	0	59	28	11	
AcHSF-A9b	2	1	3	25	12	7	
AcHSF-B1a	7	2	3	18	12	7	
AcHSF-B1b	6	5	4	18	12	10	
AcHSF-B1c	6	5	4	18	12	10	
AcHSF-B2a	6	1	4	18	10	10	
AcHSF-B2b	1	2	1	22	10	6	
AcHSF-B4b	0	0	0	14	16	11	
AcHSF-B4c	0	0	0	0	0	0	
AcHSF-C1a	3	1	3	31	28	8	
AcHSF-C1b	11	0	0	11	16	9	
AcHSF-C2	25	5	7	18	10	6	

Conserved domains and motifs of pineapple HSFs

The modular structure of the HSFs contains 5 typical conserved domains: DBD, OD, NLS, NES, and AHA domains from N to C-terminal (Table 3). The most conserved DBD domain composed of approximately 100 amino acids, containing three α-helices and a four-stranded antiparallel β-sheet (α1-β1-β2-α2-α3-β3-β4) (Fig. 4A). In addition to the DBD domain, the HR-A/B next to the DBD domain is also important and plays a crucial role in HSF-HSF interaction (Scharf et al., 2012). Besides, HR-A/B also presents in all AcHSFs (Table 3, Fig. 4B). According to the previous studies, HSFs were artificially divided into A, B, and C classes by the distinction between the HR-A and HR-B motifs (Cheng et al., 2015; Giesguth et al., 2015; Singh et al., 2012). In general, the variable length of the flexible linker between parts A and B of the HR-A/B motif of classes A and C HSFs is approximately 15 to 80 amino acids, while the HR-A/B region is tightly connected without the embedded sequence in the middle in class B members. But strangely, the insert lengths between the HR-A and HR-B have almost no difference in pineapple HSFs (Fig. 4B). And the length of the total HR-A/B domain is about 42 amino acids almost the same in pineapple classes A, B, and C HSFs, while the length of classes A and C HSFs is about 50 amino acids and 29 amino acids of class B HSFs in Arabidopsis, rice and soybean (Chauhan et al., 2011; Guo et al., 2008; Jin, Gho & Jung, 2013; Li et al., 2014).

Table 3 Functional domains of AcHSFs.

Gene	Subgroup	DBD	HR-A/B	NLS	NES	AHA motif	
AcHSF-A1a	A1	27-195	235-277	(302)KKRR	nd	(519-526)DTFWEQFL	
AcHSF-A1b	A1	26-119	159-201	(226)KKRR	nd	(443-450)DTFWEQFL	
AcHSF-A1c	A1	37-130	170-212	(238)KKRR	nd	(403-410)DSFWEQFL	
AcHSF-A2	A2	49-142	180-222	(248)KKRR	(292-299)LDLETLAL	(319-326)DEFWEELL	
AcHSF-A3	A3	113-228	263-305	(319) KEQKRIALPRPKRKFLK	nd	nd	
AcHSF-A4	A4	11-104	141-183	(202)KKRR	(136-142)HEKGLLI	(388-395)DVFWEQFL	
AcHSF-A5	A5	174-267	301-343	(371)KKRR	(327-331)LDMEQ	(558-565)DVFWEQFL	
AcHSF-A6	A6	44-137	174-216	(242)KKRR	(279-284)LDSLAL	(311-318)DGFWEELL	
AcHSF-A7a	A7	44-137	174-216	(131) KNIKRRR	nd	(305-312)EVVWEELL	
AcHSF-A7b	A7	42-130	159-197	(223)KRRR	(216-218)LLL	(271-278)DMIWEELL	
AcHSF-A9a	A9	117-210	253-295	(321)KKRR	(254-262)MQELVKLRL	(424-432)DDFDFSEQD	
AcHSF-A9b	A9	5-49	92-134	(160)KKRR	(93-101)MQELVKLRL	(263-271)DDFDFSEQD	
AcHSF-B1a	B1	17-110	165-207	(108) RRK	(252-254)VIL	nd	
AcHSF-B1b	B1	24-117	167-209	(264)DRKKGDGRKRGR	(210-219)LDVNKLDLTL	nd	
AcHSF-B1c	B1	24-117	167-209	(264)DRKKGDGRKRGR	(210-219)LDVNKLDLAL	nd	
AcHSF-B2a	B2	28-121	164-201	nd	nd	nd	
AcHSF-B2b	B2	39-132	202-244	nd	nd	nd	
AcHSF-B4b	B4	23-116	178-220	nd	(289)L	nd	
AcHSF-B4c	B4	1-47	123-165	(186)GLVDQRR	(274-279)LENEDL	nd	
AcHSF-C1a	C1	15-108	138-180	(202)KRRR	(190-192)LIL	nd	
AcHSF-C1b	C1	14-95	110-152	(166)KKKQRPGSEHKKP	(117-124)LRKEQKAL	nd	
AcHSF-C2	C2	43-136	193-235	(281)KRAR	nd	nd	
Notes.

nd, not detected.

Figure 4 Multiple sequence alignment of DNA binding domains and the HR-A/B regions (OD) of pineapple HSFs.

(A) DBD domain sequences of AcHSFs identified by Pfam database were aligned by Clustal X 2.0 software and edited by DNAMAN (v 9.0) software. The black and gray backgrounds indicate entire conservative residues and 75% conservative residues respectively. The helix-turn-helix motifs of DBD (α1-β1-β2-α2- α3-β3-β4) are shown at the bottom. Green tubes represent the α-helices and blue arrows represent the β-sheets. (B) The HR-A/B region sequences identified by SMART online tool were aligned by Clustal X 2.0 software and edited by DNAMAN (v 9.0) software. The black backgrounds indicate the 50–75% conservative residues respectively. The three line segments at the top divide HR-A core, insert and HR-B regions orderly.

The nuclear localization signals (NLS) and nuclear export signals (NES) are necessary for proteins to import and export the nucleus. The intracellular distribution of HSFs varies dynamically between the nucleus and the cytoplasm, depending on nuclear import and export balance. (Heerklotz et al., 2001; Scharf et al., 1998). After detecting, almost all the HSFs contained NLS sequences rich in basic amino acid residues (K/R), except for AcHSF-B2a, AcHSF-B2b and AcHSF-B4b. However, a total of 8 AcHSFs did not find the NES motifs. As reported in other plants, the transcription activator AHA motif was only located in class A AcHSFs, but the difference is AcHSF-A3 lacks the AHA motif (Table 3).

In addition to the typical conserved domains of HSF, we also detected the putative motifs by Multiple Em for Motif Elicitation (MEME). A total of 10 different motifs were identified in AcHSFs with lengths ranging from 20 to 50 aa (Fig. 5). The motif composition of the same group members is similar, but there are great differences among different group members. The conserved motifs in HSFs indicated that all AcHSFs contained motif 1, motif 2, except for AcHSF-A9a and AcHSF-B4c lack of motif 1. Motif 3 only exists in class A and C HSFs, not in class B. However, motif 7 only present in class A HSFs, and motif 5 only presents in class B HSFs. Additionally, some motifs were only discovered in a certain subfamily of AcHSFs, for example, motif 9 was present in the B1 subclass (Fig. 5).

Figure 5 The conserved motif analysis of 22 AcHSFs.

A total of 10 conserved motifs were identified using Multiple Em for Motif Elicitation (MEME). This is the combined match p-value. The combined match p-value is defined as the probability that a random sequence (with the same length and conforming to the background) would have position p-values such that the product is smaller or equal to the value calculated for the sequence under test.

Expression analysis of AcHSFs in different tissues

Gene expression profiles are related to their functions (Su et al., 2017). To better understand the functions of 22 pineapple AcHSF genes, the tissue-specific expression patterns were detected by 36 different tissues transcriptome sequencing, including flower (mixed stage), leaf, root, fruit S1, S2, S3, S4, S5, and S7, Se1-4, Petal 1-3, Ov 1-7, St 1-6, and Gy 1-7 from Wang et al. (2020).

The results showed some genes are highly expressed in certain tissues, while others are expressed gradually with the development of tissues (Fig. 6). For example, AcHSF-A1c and AcHSF-A7b have high expression levels in 7 fruit tissues, the expression of AcHSF-A9a gradually increased in petal development and have the highest expression value in the P3 development stage. The AcHSF-B4b and AcHSF-B4c are highly expressed in the 7 ovule development stages, which illustrate their important roles in the pineapple ovule development process. We also found that some genes showed tissue-specific expression patterns, such as the AcHSF-B2a was mainly expressed in the fruit S7 stage, AcHSF-A2 and AcHSF-A6 are highly expressed in leaf and flower tissues. In addition, the expression profiles of the genes in the same class are significantly different. For instance, three members of AcHSF-A1 have different expression patterns in all detected tissues and development stages.

Figure 6 Expression pattern of AcHSFs genes in pineapple flower and vegetative tissues.

Log2(FPKM+1) values were used for the heatmap. Se: Sepal, Pe, Petal; Ov, Ovule; Gy, Gynoecium; St, Stamen; Fr, fruit; Ro, Root; Le, Leave, Fl, Flower.

Expression profiles of AcHSFs response to various stresses

To extend our understanding of AcHSFs in response to stresses, we performed qRT-PCR to investigate the expression patterns of 6 randomly selected AcHSF genes (AcHSF-A1a, AcHSF-A2a, AcHSF-A9a, AcHSF-B2a, AcHSF-B4a, and AcHSF-C1a) in heat, cold and ABA stresses. The results illustrated that almost all of the selected AcHSFs showed similar expression patterns under the same stress conditions.

Cold stress drastically affects plant growth and development, and leads to a significant reduction in crop yield (Cai et al., 2015); therefore, plants must respond quickly to cold stress. As shown in the results, under the cold stress treatment, the expression of all the 6 AcHSFs increased rapidly from 0 h to 24 h and then reduced at 48 h (Fig. 7A). These may indicate that AcHSFs are commonly up-regulated within a short-timer by cold stresses, and then the expression is down-regulated rapidly. Heat shock transcription factors play crucial roles in response to heat shock induction. The result showed that the expression of 6 AcHSFs continues to increase from 0 h to 48 h in heat stress treatment (Fig. 7B). After ABA treatment, the expression of most selected AcHSF genes increased from 0 h to 12 h, and then decreased after 12 h, while the expression of AcHSF-A2a continued to increase (Fig. 7C). This result implies that the expressions of AcHSFs were suppressed under the longtime ABA treatment and might play crucial roles in different stress (cold and heat, etc.) response pathways.

Figure 7 qRT-PCR expression analysis of 6 selected AcHSF genes in response to different abiotic stress treatments.

(A) Cold stress treatment (4 °C) (B) Heat stress treatment (45 °C); (C) 100 nM ABA treatment. Mean expression value was calculated from three independent replicates. Error bars indicate ±SD (n = 3). Asterisks on top of the bars indicating statistically significant differences between the stress and counterpart controls (∗P < 0.05, ∗∗P < 0.01).

Discussion

During its growth and developmental stages, the pineapple is severely destroyed by various abiotic stresses (cold, heat, drought, etc.) and biotic stresses (especially fungal pathogen infection). HSFs are among the critical regulatory components of various abiotic and biotic stresses in plants. This research identified and characterized, for the first time, a systematic genome-wide review of the AcHSF family. Consequently, from the pineapple genome, a total of 30 AcHSF genes were identified. The widely accepted model of HSFs defines the necessity of HSF-type DBD and OD characterized by a coiled-coil structure. Thus, due to the absence of HSF-type DBD domains and/or coiled-coil structures, 8 of them were discarded. Meanwhile, pineapple HSF has a similar subfamily distribution compared with the monocots plant O. sativa, but is different from dicots plants A. thaliana and P. trichocarpa. Some genes are unique to monocots or dicots. For example, the subclasses AcHSF-A8 and AcHSF-B3 are confined to dicots, while AcHSF-A9 and AcHSF-C2 are characteristic of monocots, suggesting that different evolutionary events of HSF genes occurred in dicots and monocots (Fig. 2, Table 1).

Research on gene expression regulation mediated by introns has made significant progress (Le Hir, Nott & Moore, 2003; Li et al., 2019; Rose, 2008; Shaul, 2017). Therefore the study of gene structure is beneficial to elucidate the gene function. Analysis of AcHSFs gene structures revealed that most of the classes A AcHSFs contain more than one intron, and several AcHSFs have 3 or 4 introns, such as AcHSF-A1a, AcHSF-A1b, and AcHSF-A9a. However, the genes in the class B and C only contain 1 intron, except for AcHSF-C1b (Fig. 3). This particular intron structure may be due to the specific functions of the AcHSF genes. The required DBD domain and unique protein domains (HR-A/B, NLS, NES, RD, and AHA) are found in all 22 AcHSF proteins (Table 3, Fig. 4), which provide the structural basis for their conserved function (Giorno et al., 2012). The HSF DBD domain contains approximately 100 amino acids and is strongly preserved in various plant-to-animal species; we also found the same conserved domain in pineapple (Fig. 4A). As reported previously in other plants, the transcription activator AHA motif was only located in class A AcHSFs, but AcHSF-A3 lacks the AHA motif (Table 3). The HSFs that lack AHA domains might contribute to the activator’s function differently or form hetero-oligomers by binding to other HSF-As (Guo et al., 2008).

The expression patterns analysis of different AcHSFs showed that AcHSF-B4b and AcHSF-B4c are highly expressed in 7 ovule development stages, indicating the potential functions in pineapple ovule development. The high expression levels of AcHSF-A7b, AcHSF-C2, and AcHSF-A1c in fruit development stages uncovered their important roles in fruit development (Fig. 6). Furthermore, we found that the expression of AcHSF-A9a gradually increased throughout the development stage and reached the highest expression level in the third stage of petal development. AcHSF-A2 and AcHSF-A6 have high expression levels in leave and mixed flower tissues (Fig. 6). These data suggest that AcHSFs may regulate several developmental processes. The stress response is very significant for plant growth and development. Previous studies have shown that the HSF genes are involved in several abiotic stress response, including heat, cold, drought, and salt stress responses in different plants such as Arabidopsis, tomato, apple, Populus euphratica, and Phyllostachys edulis (Fragkostefanakis et al., 2015; Giorno et al., 2012; Ikeda, Mitsuda & Ohme-Takagi, 2011; Xie et al., 2019; Xue et al., 2014; Zhang et al., 2016a). In our study, most of the selected AcHSFs showed similar expression patterns under the same stress conditions. Under the cold stress (4 °C) treatment, the expressions of AcHSFs were induced from 0 h to 12 h and then inhibited after 12 h (Fig. 7A). The same expression pattern was also observed in the 100 mM ABA treatment, but the difference was that the AcHSFs were more sensitive to ABA treatment (Fig. 7C). The continuous increase in the expression pattern of AcHSFs was observed at 45 °C treatment, indicating that heat stress-induced the expression of AcHSFs (Fig. 7C).

Taken together, this study is the first to identify the AcHSF family genes, their properties as well as their expression profiles. This information could be used to utilize them as potential candidates in a breeding program of pineapple. However, gene expression and function analysis are complicated biological mechanisms, and additional studies are necessary to interpret the regulatory process.

Conclusions

In the present study, 22 AcHSF genes were identified in pineapple (Ananas comosus) and generated detailed information on the gene and protein structures. The expression profiles of various tissues and developmental stages were analyzed by the RNA-seq data, which may help to study their functions in different developmental processes or regulatory pathways. We also showed that some AcHSF genes participate in various biotic and abiotic stresses (heat, cold, and ABA), which may help develop new pineapple varieties with desired agronomic traits stress tolerance.

Supplemental Information

Supplemental Information 1 The protein sequences of pineapple HSFs

Click here for additional data file.

Supplemental Information 2 The protein sequences of rice HSFs

Click here for additional data file.

Supplemental Information 3 The protein sequences of Arabidopsis HSFs

Click here for additional data file.

Supplemental Information 4 The protein sequences of Populus trichocarpa HSFs

Click here for additional data file.

Supplemental Information 5 The promoter sequences of Pineapple HSFs

Click here for additional data file.

Supplemental Information 6 The cis-elements analysis in Pineapple HSFs promoter sequences

Click here for additional data file.

Supplemental Information 7 The expression profiles of the pineapple HSF genes in different tissues

Click here for additional data file.

Supplemental Information 8 Primers used for qRT-PCR of AcHSFs genes

Click here for additional data file.

Supplemental Information 9 qRT-PCR results of the pineapple HSF verification genes

Click here for additional data file.

We would like to thank the reviewers for their helpful comments on the original manuscript.

Additional Information and Declarations

Competing Interests

Author Contributions

DNA Deposition

Data Availability

The authors declare there are no competing interests.

Lulu Wang conceived and designed the experiments, performed the experiments, analyzed the data, prepared figures and/or tables, authored or reviewed drafts of the paper, and approved the final draft.

Yanhui Liu performed the experiments, prepared figures and/or tables, and approved the final draft.

Mengnan Chai and Huihuang Chen analyzed the data, prepared figures and/or tables, and approved the final draft.

Mohammad Aslam, Yuan Qin and Hanyang Cai conceived and designed the experiments, authored or reviewed drafts of the paper, and approved the final draft.

Xiaoping Niu analyzed the data, prepared figures and/or tables, authored or reviewed drafts of the paper, and approved the final draft.

The following information was supplied regarding the deposition of DNA sequences:

The RNA-seq data of ovule, stamen, petal, Gynoecium, and Sepal were downloaded in the European Nucleotide Archive (ENA): PRJEB38680.

The RNA-seq datasets of Root, Leaf, Flower, and Fruit were downloaded from http://pineapple.angiosperms.org/pineapple/html/download.html.

The following information was supplied regarding data availability:

The raw measurements showing all the analysis processes and results are available in the Supplementary Files.

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
