# Peer review of "Genome-wide identification, classification, and expression analysis of the HSF gene family in pineapple (Ananas comosus)"

_PeerJ, doi:10.7717/peerj.11329_

## Round 0.1 · original submission · Minor Revisions

Dear Dr. Qin,

Your manuscript, which you submitted to PeerJ, has been reviewed by two experts in your research area. The comments from reviewers are included at the bottom of this letter and in the attachments.

Both reviewers comment on the important and good work but point out that the manuscript needs significant improvement. Also, I encourage you to get with an English-speaking professional person to edit your revision to avoid some jargon and more clear.

My review copy is also attached for your information. Thank you again for submitting your manuscript to PeerJ for publication.

Best regards,

Sincerely,

Tika Adhikari

Reviewer 1 ·

Basic reporting

The article includes sufficient introduction and background to demonstrate how the work fits into the broader field of knowledge.

Experimental design

Research question well defined, relevant and meaningful. The study also opens several important questions that probably the authors will address in the future.

Validity of the findings

Conclusions are well stated and linked to original research question.

Additional comments

The manuscript entitled “Genome-Wide Identification, Classification, and Expression Analysis of the HSF Gene Family in pineapple (Ananas comosus)” reports about identification of 22 AcHSF genes in pineapple, analysis of the gene and protein structures, as well as HSF functions in various developmental processes or regulatory pathways. The manuscript was easy to read and in general is well described. However, I have several comments about the manuscript:

L48-49. ….act as repressors of the expression of heat-inducible… It seems like the word was lost (after heat-inducible).
L90. This sentence should be corrected (to obtained…). At the same time, it remains unclear which plant DBD domain was used for screening.
L96. What does it mean the Chromosome NO.? Probably it means location on the chromosome. This phrase should be modified.
L104. The same comment.
L117. Authors described the gene. The AcHSFs should be written in italics.
L124. Conserved domains and motifs analysis were performed for proteins. It means that AcHSFs should not be written in italics in this case.
L136-137. The transcriptomic data of pineapple in different organs and developmental stages……… This phrase should be corrected.
L161. Please check the name of this section.
L282. The resulted… (probably, the results).
L308. …..in respond to the induction of heat shock. Probably, ….in response to the induction of heat shock.
L350. … to other HSFAs. Please present names in a universal form such as HSF-A in all text.
L361. The HSF should be written in italics. The tomato and apple should not be written in italics.

Annotated reviews are not available for download in order to protect the identity of reviewers who chose to remain anonymous.

·

Basic reporting

In the submitted manuscript the authors give a comprehensive description of the HSF transcription factor family in Ananas and their expression characteristics.

The provided background information are sufficient. Most of the tables and figures are in a self-explanatory form, but some of them would need some improvement. In Fig. 5 p-values are given, but there is no explanation for what they have been assigned to. In Table 3 the acronym nd has not been explained. For Table 4 I must admit that it does not make sense to me. It seems that in this table the different motifs found in HSFs have been compared with each other. The sense of this comparison is not apparent and does not contribute any further understanding of the protein family. Therefore, I would suggest to delete it.

The language used in this manuscript needs thorough correction. In some cases words are missing (like in line 45, 182) or they are wrong (like line 48, 225, Table 1 "genome" instead of "genomics"). But there are a lot of passages which are misunderstandable (e.g. lines 190-200; see also section 3). Please give the text to a native english speaker for thorough correction.

Experimental design

The experimental design of the presented study meets the criteria defined by the journal. Therefore I have no comments, except for the description of the promotor sequences which seems to be rather unclear. According to the information given in lines 190-200 there were no upstream sequence information available for 14 HSFs. But for three of them (A4, C1b and B4c, lines 211-213) a lack of elements have been described for which no sequence information available. Please check this part.

Validity of the findings

No comment.

Additional comments

As stated above I would strongly recommend thorough text editing.

---

## Round 0.2 · Minor Revisions

Dear Dr. Qin,

Thank you for submitting your revised manuscript to PeerJ. Your manuscript has been accepted for publication. However, it has brought to my attention, there were some comments/suggestions from the Section Editor (please see below). I would appreciate it if you could incorporate it in your revision. Thank you.

"Ming (2015) should be cited for the genome sequence, not just expression data. +++ Figure 6 (heatmap) is poorly constructed. Having a two-color heatmap (blue and red) only makes sense when doing a differential expression analysis, then one color indicates up-regulation, and one color regulated down-regulation (and white or black indicates no change). In this case, since there is no differential expression analysis, a single gradient should be used, from white (or black) indicating no expression to more and more colored with higher expression."

Congratulations.

Sincerely,

Tika Adhikari

---

## Round 0.3 · Minor Revisions

Dr. Qin,

This letter is to follow up on my previous comment regarding the paper you cited (Wang 2020). Both Section Editor and I believe that this (Wang et al. 2020) is related to a transcriptome paper. Apparently, your current manuscript under consideration for publication in PeerJ relies mainly on the genome paper described by Ming (2015). Therefore, I request you to revise/replace this reference Wang et al. (2015) with Ming (2015) at your earliest possible time. Thank you.

Best regards,

Sincerely,

Tika Adhikari

---

## Round 0.4 · accepted · Accept

Greetings.

I am writing to inform you that your manuscript - Genome-wide identification, classification, and expression analysis of the HSF gene family in pineapple (Ananas comosus) - has been accepted for publication.

Please use the attached pdf file for further editing.

Best regards,

Tika